# Preexisting Virus-Specific T Lymphocytes-Mediated Enhancement of Adenovirus Infections to Human Blood CD14+ Cells

**DOI:** 10.3390/v11020154

**Published:** 2019-02-13

**Authors:** Fengling Feng, Jin Zhao, Pingchao Li, Ruiting Li, Ling Chen, Caijun Sun

**Affiliations:** 1School of Life Sciences, University of Science and Technology of China (USTC), Hefei 230027, China; Feng_fengling@gibh.ac.cn; 2School of Public Health (Shenzhen), Sun Yat-sen University, Guangdong 518107, China; zhaoj47@mail2.sysu.edu.cn (J.Z.); lirt9@mail2.sysu.edu.cn (R.L.); 3State Key Laboratory of Respiratory Disease, Guangzhou Institutes of Biomedicine and Health (GIBH), Chinese Academy of Sciences, Guangzhou 510530, China; li_pingchao@gibh.ac.cn

**Keywords:** Adenovirus, T lymphocytes, Monocytes, Cytokines, Pathogenesis

## Abstract

Antigen-specific T lymphocytes play a critical role in controlling viral infections. However, we report here that preexisting virus-specific T cell responses also contribute to promoting adenovirus (Ad) infection. Previously, we found that CD14+ monocytes from Ad-seropositive individuals exhibited an increased susceptibility to Ad infection, when compared with that of Ad-seronegative individuals. But the underlying mechanisms for this enhancement of viral infection are not completely clarified. In this study, we found that the efficacy of Ad infection into CD14+ monocytes was significantly decreased after CD3+ T lymphocytes depletion from PBMC samples of Ad-seropositive individuals. In contrast, adding virus-specific CD3+ T lymphocytes into PBMC samples of Ad-seronegative individuals resulted in a significant increase of infection efficacy. CD3+ T lymphocytes in PBMC samples from Ad-seropositive individuals were more sensitive to be activated by adenovirus stimulus, characterized by upregulation of multiple cytokines and activation markers and also enhancement of cell proliferation. Further studies demonstrated that GM-CSF and IL-4 can promote Ad infection by up-regulating the expression of scavenger receptor 1 (SR-A) and integrins αVβ5 receptor of CD14+ cells. And taken together, these results suggest a novel role of virus-specific T cells in mediating enhancement of viral infection, and provide insights to understand the pathogenesis and complicated interactions between viruses and host immune cells.

## 1. Introduction

When organism is attacked by external microorganisms, immune responses are elicited to fight against these infections, which lead to subtle interactions between host immune cells and microorganisms. Compared with innate immune responses, adaptive immune responses are highly specialized, rapid and powerful in response to a particular pathogen. Adaptive immune responses are mainly involved with B lymphocytes-mediated humoral immunity and T lymphocytes-mediated cellular immunity. It is well known that both B cells (by secreting antibodies) and T cells (by producing cytokines and chemokines) play critical roles to prevent and eliminate pathogens [1,2,3]. With the long-term struggle between viruses and their hosts, however, many viruses have evolved kinds of mechanisms to “turn enemies into allies”, thereby hijacking and converting the immune system for facilitating their entry, replication and spread [4,5,6,7]. For example, antibody is usually to be thought as an antiviral factor, and it can recognize, bind and kill foreign pathogens by different protective mechanisms, including agglutination, activation of complement, opsonization, antibody-dependent cell-mediate cytotoxicity (ADCC) and neutralization [8]. However, under certain conditions, viruses can bind to some antibodies to form antibody-virus complex to enhance their infectivity to host cells. This phenomenon is termed as antibody-dependent enhancement (ADE). ADE phenomenon has been reported in kinds of viruses and parasite, such as human immunodeficiency virus (HIV), dengue fever virus (DENV), Zika Virus, adenovirus (Ad) and malaria, etc. [9,10,11]. One explanation for ADE is that antibody-virus complex is bound to the Fc receptor on the surface of host cells, and then internalized into cells by phagocytosis [12]. Although some viruses can exploit B cell-mediated antibody response, there is no report whether viruses can exploit T cell-mediated cellular response to enhance their infection.

Human adenoviruses group C, including serotype 2 (Ad2) and serotype 5 (Ad5), are known to be a common cause of respiratory tract infection. Our previous work and others studies demonstrated that there was a high prevalence of anti-Ad2 and anti-Ad5 neutralizing antibodies in worldwide populations, implying ubiquitously prior natural infection [13]. The virology, epidemiology and cell biology of Ad2/Ad5 have been well characterized over the past decades, and they have been extensively developed as popular vectors for gene delivery due to their ability to infect a broad tropism of host cells without integration into the host genome, accommodate a large fragment of foreign transgenes, and culture in high yield in vitro [14]. Recombinant Ad2 and Ad5 have been explored for immunotherapy, gene therapy and vaccination for cancer and infectious diseases [15,16,17]. Ad5-vectored P53 and oncolytic Ad5 have been approved for cancer treatment in clinic use [18], and Ad5-based Ebola vaccine has been tested in clinic trials [19,20]. There is a good safety record from many clinical trials of Ad-vectored products, but it is noted that the symptoms caused by Ad2/Ad5 infection range from the common cold to pneumonia, even leading to severe life-threatening disease in some immunocompromised subjects [21]. In view of ubiquitously prior natural infection of Ad2/Ad5 and extensive administration of Ad5-vectored products in huge population [22,23], it is very important to further clarify its potential pathogenesis and complicated interactions between host immune cells and adenovirus infection.

Previously, we reported that there was an increased infectivity for Ad5 to CD14+ monocytes of Ad5-seropositive subjects, when compared with that of Ad5-seronegative subjects [24]. In addition, we recently reported that the upregulated macrophage scavenger receptor 1 (SR-A) expression on the surface of CD14+ monocytes partially contributed to this enhanced entry of Ad5 [25]. But the underlying mechanisms are not completely clarified for this increased infectivity of Ad5. In this study, we further report that preexisting virus-specific CD3+ T lymphocytes also contribute to promoting Ad infection into monocytes/macrophages by secreting GM-CSF and IL-4 cytokines, which upregulated the expression of SR-A and integrins αVβ5 receptor. The results provide insights to understand the complicated interactions between host immune cells and viral infection.

## 2. Materials and Methods

### 2.1. Viruses, Cells and Ethic Statement

Recombinant adenoviruses, including Ad5-EGFP (enhanced green fluorescent protein), Ad5-SEAP (secreted alkaline phosphatase), Ad2-EGFP and Ad2-SEAP were used in this study and generated according to our previously reported methods [24,26]. Those recombinant Ads are replication-defective due to E1 gene/E3 gene-deletion. The inserted reporter gene in recombinant Ad is driven by CMV promotor. The titers for Ad5-EGFP, Ad5-SEAP, Ad2-EGFP and Ad2-SEAP are 6.8 × 10^12^ vp/mL, 4.13 × 10^12^ vp/mL, 6.6 × 10^12^ vp/mL and 5.4 × 10^12^ vp/mL respectively.

Fresh peripheral blood mononuclear cells (PBMCs) were isolated from human volunteers, and cultured at 37 °C in 5% CO_2_ incubator in RPMI 1640 medium containing 10% FBS (Gibco, New York, NY, USA), 2 mM l-glutamine (Gibco), 1 mM sodium pyruvate (Gibco), 10 mM HEPES (Gibco) and 55 µM β-mercaptoethanol (Gibco).

The experimental protocol of blood samples involved with human volunteers was approved (Permit Number: 2009039) by our institutional Ethics Committee of the Guangzhou Institutes of Biomedicine and Health (GIBH), Chinese Academy of Sciences. Whole blood and EDTA-anticoagulant blood samples were taken from healthy participates with written informed consents. The whole blood was used for serum isolation for antibody and cytokine detection, and the EDTA-anticoagulant blood was used for PBMCs isolation for subsequent culture and FACS detection.

### 2.2. PBMC Isolation, Adenovirus Infection and Analysis for Infectivity

Human PBMC cells were obtained by density gradient centrifugation, using OptiPrep™ lymphocyte separation solution (Axis Shield Poc As, Rodeløkka, Oslo, Norway) following the manufacturer’s directions. After counting, PBMCs were incubated with the indicated dosage of the above adenoviruses for 1 h at 1000× *g* centrifugation, and then cultured for 24–48 h at 37 °C in 5% CO_2_ incubator. For detecting the expression of EGFP reporter gene in different cell population, the infected PBMCs were incubated with corresponding fluorescent-labeled monoclonal antibodies (CD3-APC, CD3-PE, CD3-PerCP, CD14-APC, CD14-PE, CD19-PE-cy5, CD56-PE, CD27-APC, CD95-PE, HLADR-APC, Ki67-PE, 7-AAD, BD Pharmingen, San Diego, CA, USA) and CD38-FITC (STEMCELL Technologies, Vancouver, Canada), Integrinβ5-PE (eBioscience, San Diego, CA, USA), and then detected with a BD FACS LSR Fortessa flow cytometer (BD Biosciences, San Diego, CA, USA).

For detecting the expression of SEAP reporter gene, PBMCs were seeded at 5 × 10^5^ cells per well in 96-well plates, and then incubated with the indicated dosage of Ad-SEAP for 24–48 h at 37 °C in 5% CO2 incubator. A total of 50 μL cell-free supernatant was taken from each sample to detect SEAP activity using a Phospha-Light kit (Applied Biosystems, Foster City, CA, USA). Relative light units (RLU) were monitored in a luminometer (MLX Microtiter, Dynex Technologies, Inc., Chantilly, VA, USA).

### 2.3. Sorting of Different Cell Subsets to Detect the Infectivity for Adenovirus

CD3+ T lymphocytes and CD19+ B lymphocytes were separated from PBMCs by magnetic bead-based cell sorting kit (MACS, Miltenyi Biotec, Bergisch Gladbach, Germany), following the manufacturer’s directions. In brief, purified PBMCs were washed with sorting buffer and then incubated with corresponding magnetic bead-labeled monoclonal antibodies at 4 °C for 15 min. After washing and suspension, the labeled cells were added to autoMACS Pro Separator (Miltenyi Biotec, Bergisch Gladbach, Germany). The unlabeled negative fraction and labeled positive fraction were collected respectively for FACS analysis and infection experiment as described above.

### 2.4. Quantitative PCR

Total mRNA from different cell samples was isolated using QIAGEN RNeasy Protect Mini Kit (Cat No:74126, Hilden, Germany), and then the concentration of mRNA was detected with NanoDrop 8000 (Thermo, Waltham, MA, USA) and all the sample was adjusted to the same concentration. The mRNA was served as templates for the quantitative PCR. Quantitative PCR was carried out with CFX96 Touch (Biorad, Hercules, CA, USA) with QuantiFast SYBR Green RT-PCR Kit (Cat No:204057, QIAGEN, Germany,). Cycle threshold (C(t)) values and melting curves were analyzed with Bio-Rad CFX manager 3.1 as our previously reported [24,25]. The relative numbers of desired molecular, including CAR, integrin alpha v beta 5 (αvβ5), interferon (IFN)-γ, granulocyte macrophage-colony stimulating factor (GM-CSF), interleukin (IL)-4, etc., were determined by comparison with the level of beta actin copies. The primer sequences used in this study are available in Appendix A. The final data are represented as the mean values of triplicate tests.

### 2.5. Assay for SEAP-Based Ad Neutralizing Antibody

Specific Ad2 and Ad5 neutralizing antibody titers were quantitatively determined as our previously reported methods [23,27].

### 2.6. IFN-γ ELISPOT Assays

IFN-γ ELISPOT assays for adenovirus-specific T cell responses were conducted following our previously reported protocol [26,28] with minor modifications. In brief, anti-IFN-γ monoclonal antibody-coated 96-well plates (Millipore, Immobilon-P membrane, Burlington, MA, USA) were added with 4 × 10^5^ PBMCs with or without the lysed adenovirus particles as antigen stimulus (2 μg/mL), and 10 µg/mL concanavalin A (Sigma-Aldrich, St. Louis, MO, USA) was used as a positive control. After incubated for 24 h in 5% CO_2_ incubator, the plate was washed and incubated with biotinylated anti-IFN-γ detection antibody (U-Cytech) at 4 °C overnight. At last, spots were developed by incubating in NBT/BCIP substrate (Pierce, Rockford, IL, USA), and counted with ELISPOT reader (Bioreader 4000). Data are showed as the quantity of spot-forming cells (SFC) per million cells.

### 2.7. Incubation with Cytokines during Adenovirus Infection

To detect the direct effect on adenoviral infection by cytokines, PBMC was seeded at 5 × 10^5^ cells per well in 96-well plates, and then infected with Ad-EGFP or Ad-SEAP at 1250 vp/cell. Meanwhile, recombinant Human GM-CSF, IL-4 or IFN-γ (R&D system, Minneapolis, MN, USA) was added respectively at final concentration of 20 ng/mL, 20 ng/mL, 40 ng/mL. These cells were incubated for 24–48 h at 37 °C in 5% CO_2_ incubator. The expression level of adenovirus receptor and the efficacy of adenovirus infection were detected as descripted above.

### 2.8. Blockade with Anti-Cytokines Antibodies during Adenovirus Infection

To further verify the roles of GM-CSF and IL-4 in promoting adenovirus infection, PBMC was seeded at 2 × 10^5^ per well in 96-well plates or 1 × 10^6^ cells per well in 24-well plates, and then infected with Ad5-SEAP or Ad5-EGFP at 1250 vp/cell. To block the possible function of cytokines, anti-human GM-CSF monoclonal antibody (clone: BVD2-21C11, Biolegend, San Diego, CA, USA) or anti-human IL-4 monoclonal antibody (clone: 8D4-8, Biolegend, USA) or matched isotype-control antibody was simultaneously added to the culture in different concentrations. Then, these cells were incubated for 24 h at 37 °C in 5% CO_2_ incubator. The efficacy of adenovirus infection was detected as descripted above.

### 2.9. Data Analysis

FACS data were analyzed using FlowJo v10 software (Tree Star, Inc., Ashland, OR, USA). For statistical tests, two-tailed *p* value was performed with GraphPrism 5.01 (GraphPad software Inc., LaJolla, CA, USA).

## 3. Results

### 3.1. CD14+ Monocytes from Ad-Seropositive Individuals Exhibited an Increased Susceptibility to Ad Infection

In our previous studies, it was observed that CD14+ cells from Ad5-seropositive subjects exhibited an increased susceptibility to Ad5 entry, than those from Ad5-seronegative subjects [24,25]. To further confirm this phenomenon, we performed the present study with different blood samples. Consistent with previous methods, we used firstly Ad5-EGFP as a reporter virus to represent the efficacy of Ad5 infection. The gate strategy is showed in Figure 1A, and 7-AAD is used to exclude the dead cells. 23 healthy volunteers, including Ad5-seronegtive (Nab < 18, *n* = 8) and Ad5-seropositive (Nab titer > 18, *n* = 15) people, were involved in this experiment. When comparted with that of Ad5-seronegative samples, both the percentage of EGFP-positive cells (Figure 1B) and expression level of EGFP protein (mean fluorescent intensity, MFI, Figure 1C) were significantly higher in CD14+ cells of Ad5-seropositive people (*p* < 0.0001). Moreover, a similar phenomenon was expanded to the efficacy of Ad2 infection. Another 22 healthy volunteers, including Ad2-seronegtive (Nab < 18, *n* = 7) and Ad2-seropositive (Nab titer > 18, *n* = 15) people, were involved in this experiment. CD14+ cells from Ad2-seropositive people (*n* = 15) were more susceptible to Ad2-mediated EGFP expression than that of Ad2-seronegative people (*n* = 7) (Figure 1D and 1E). In addition, the alternative SEAP reporter gene-based adenoviruses were administrated to further verify this observation. After incubating with Ad5-SEAP (Figure 1F) or Ad2-SEAP (Figure 1G), there was a significantly stronger expression level of relative light units (RLU) in samples from Ad-seropositive people in a dose-dependent manner. Those results confirmed that CD14+ monocytes from Ad-seropositive subjects exhibited an increased susceptibility to Ad2 and Ad5 infection.

### 3.2. Secretory Factors by PBMC Samples of Ad-Seropositive Individuals Contributed to Promoting the Efficacy of Ad Infection

Next, we attempted to elucidate the exact mechanism for this enhanced adenovirus infection for Ad5-seropositive samples. The elevated receptors on the surface of targeted cells, such as scavenger receptor class A (SR-A) and integrin [24,25], might be partially associated with this observation, but the underlying mechanism is not fully understood. To further elucidate if there are other factors (such as secretory cytokines by activated lymphocytes) to affect the efficacy of Ad infection, we performed this experiment of mixed lymphocyte culture. Three paired PBMCs from Ad5-seronegative subjects and Ad5-seropositive subjects were mixed at each dilution ratio (Figure 2A), and then infected by Ad5-EGFP. Besides of the receptors (such as CAR, integrin and SR-A) on the surface of targeted cells, we speculated if there were no other secretory factors to affect the efficacy of Ad infection, the predicted values of the different ratios of PBMCs mixture were expected to be a linear relation. However, the actual values of infection efficacy in our study were a nonlinear curve relation, and significantly higher than the calculated values (Figure 2B,C), implying that the secretory cytokines by activated lymphocytes did affect the efficacy of Ad infection.

Considering that preexisting virus-specific T cells of PBMCs from Ad-seropositive individuals were expected to be rapidly activated to secrete soluble cytokines in response to Ad re-infection, we speculated that these cytokines by activated T cells can act on other cells around them, and further convert some CD14+ cells of Ad-seronegative samples to be more susceptible to Ad5 entry. In the meantime, T cells are usually activated when mixing the allogeneic leukocytes from different donors. Those data suggested that some soluble factors beyond targeted cell self might play a crucial role for this enhanced infectivity.

### 3.3. The Efficacy of Adenovirus Infection into CD14+ Monocytes were Significantly Affected by Removing or Adding CD3+ T Lymphocytes from PBMC Samples of Ad-Seropositive Individuals

It is well known that the mainly targeted cells in blood for Ad5 infection are CD14+ monocytes/macrophages, and other cell populations (such as T lymphocytes, B lymphocytes) are poorly susceptible to Ad5 infection. However, it is not clear whether T lymphocytes and/or B lymphocytes indirectly affect the efficacy of adenovirus infection into CD14+ cells. To dissect this issue, CD3+ T lymphocytes and CD19+ B lymphocytes were magnetically sorted from fresh PBMCs as described in methods, and then the efficacy of Ad infection was detected. Then, the purity of sorted cells and the percentage of residue contaminating cells was determined by FACS (Figure 3A and Appendix A). In this study, the purity of sorted CD3+ T cells and CD19+ B cells was 90.02%–97.2% and 76.23%–80.02% respectively. The residue contaminating B cells in sorted CD3+ T cells was 0.54%–1.63%, and the residue contaminating T cells in sorted CD19+ B cells was 11.10%–15.02%. The percentage of CD19+ B cells in sorted CD3-/CD19- sample was 0.23%–0.55%, and the percentage of CD3+ T cells in sorted CD3-/CD19- sample was 0.11%–0.38%. Interestingly, we found that the efficacy of adenovirus infection into CD14+ monocytes was significantly decreased after removing CD3+ T lymphocytes from PBMC samples of Ad-seropositive individuals (preexisting virus-specific CD3+ T cells). Moreover, this decrease was restored when adding CD3+ T lymphocytes from Ad-seropositive individuals, as indicated by both the percentage of EGFP-expressing cells (Figure 3B) and mean fluorescent intensity (MFI) of EGFP level (Figure 3D). In our study, adding CD3+ T lymphocytes from Ad-seronegative individuals (non-specific CD3+ T cells) did not significantly enhance the efficacy of Ad infection, although there is a trendy of partially restored. This partial restoring might be caused by the non-specific cell activation by the positive selection via magnetic microbeads. The efficacy of adenovirus infection had been scarcely changed, when removing CD3+ T lymphocytes from PBMC samples of Ad-seronegative individuals. In addition, adding virus-specific CD3+ T cells into PBMCs of Ad-seronegative individuals resulted in a significant increase of infection efficacy (Figure 3C and 3E). Those data demonstrated that virus-specific CD3+ T lymphocytes played a crucial role to facilitate the efficacy of adenovirus infection into CD14+ cells.

### 3.4. Preexisting Virus-Specific CD3+ T Lymphocytes were More Effectively Activated after Encountering Adenovirus Stimulus

We and other groups reported previously that adenovirus infection leaded to a stronger immune activation in monocytes of Ad5-seropositive subjects when compared with that of Ad5-negative subjects [24,25]. In this study, we further detected how adenovirus infection interacted with preexisting virus-specific CD3+ T lymphocytes from Ad5-seropositive subjects. Ad5-specific T lymphocytes in Ad5-seropositive PBMCs were stimulated with lysed adenovirus particles, and then detected by IFN-γ-mediated ELISPOT assay. Results demonstrated that the frequency of Ad5-specific IFN-γ-secreting T lymphocytes was significantly higher than that of T lymphocytes in Ad5-seronegative PBMCs (Figure 4A, *p* < 0.0001), implying that there was high level of preexisting Ad5-specific T memory cells in Ad5-seropositive subjects. Then, the level of proliferation and activation of T lymphocytes were detected, and we found that the cell proliferation marker (Ki67, Figure 4B) and the activating markers, including HLA-DR, CD38, CD95 and CD27 (Figure 4C–F), in T lymphocytes from Ad5-seropositive subjects were significantly upregulated when compared with that of Ad5-negative subjects. A consequence of the promoted proliferation and activation of memory T lymphocytes is to increase the production of cytokines/chemokines. In our previous study [24,25], it was reported that the secretions of a panel of cytokines/chemokines were significantly unregulated in the supernatant of PBMCs from Ad5-seropositive subjects, when stimulated by Ad5 infection. Herein, we confirmed that the production of cytokines/chemokines, including interferon (IFN)-γ (Figure 4G), granulocyte macrophage-colony stimulating factor (GM-CSF) (Figure 4H) and interleukin (IL)-4 (Figure 4I), was more effectively elicited in T lymphocytes from Ad5-seropositive subjects, when compared with that of Ad5-seronegative subjects. Those data demonstrated that preexisting Ad5-specific T lymphocytes were more effectively activated after encountering adenovirus infection.

### 3.5. Incubation of GM-CSF and IL-4 Promoted the Efficacy of Adenovirus Infection by Upregulating SR-A and Integrin Receptor

Next, we investigated how these elevated cytokines affect the efficacy of adenovirus infection. Ad-EGFP virus was added into freshly isolated PBMCs with or without recombinant IFN-γ, GM-CSF or IL-4 respectively, and the EGFP expression was detected after 48 h. When comparted with mock-treated samples, both the percentage of EGFP-positive cells (Figure 5A) and expression level of EGFP protein (MFI, Figure 5B) were significantly increased in CD14+ cells of GM-CSF/IL-4-treated samples. However, treatment with IFN-γ decreased significantly the efficacy of Ad infection, since IFN-γ is a well-known antiviral factor. Moreover, Ad-SEAP reporter virus was used to further confirm these results. After Ad-SEAP incubation with or without IFN-γ, GM-CSF or IL-4 respectively, the expression level of SEAP was significantly enhanced in GM-CSF/IL-4-treated samples but inhibited in IFN-γ-treated samples (Figure 5C).

To further verify the roles of GM-CSF and IL-4 in promoting adenovirus infection, we detected whether the function blockade of GM-CSF and IL-4 would cripple the adenovirus infection. As expected, both the frequency of EGFP-positive cells (Figure 5D) and expression level of EGFP protein (MFI, Figure 5E) were significantly decreased in CD14+ cells of anti-human GM-CSF monoclonal antibody-treated samples in a dose-dependent manner, when comparted with mock-treated samples. Moreover, after Ad-SEAP incubation with anti-human GM-CSF monoclonal antibody simultaneously, the expression level of SEAP was significantly reduced in a dose-dependent manner (Figure 5F). In addition, the similar results were found when Ad-EGFP or Ad-SEAP was incubated with anti-human IL-4 monoclonal antibody simultaneously (Figure 5G–I).

Furthermore, to elucidate the underlying mechanism of the increased susceptibility to Ad infection by GM-CSF/IL-4 treatment, we subsequently detected the expression level of adenovirus-related receptors by quantitative RT-PCR, such as CAR, αVβ5 integrin and SR-A. Results demonstrated that SR-A and αVβ5 integrin were significantly upregulated by treatment with IFN-γ, GM-CSF and IL-4 (Figure 5J,K), while the expression of CAR receptor was not changed (data not shown). In addition, the integrin receptor protein on the cell surface was analyzed by flow cytometer. Consistent with above data, the integrin on the cell surface was significantly upregulated by treatment with IFN-γ, GM-CSF and IL-4 (Figure 5L). Therefore, those data demonstrated that GM-CSF/IL-4-treated CD14+ monocytes increased the susceptibility to Ad infection through upregulating the expression of SR-A and integrin receptor.

## 4. Discussion

The immune system is developed as a host defense system against a variety of diseases, especially infectious diseases. After long-term interactions between pathogens and their hosts, multiple defense functions of immune system have been developed to recognize and eliminate pathogens, like antigen-specific T lymphocytes are important for controlling viral infections. *Vice versa*, pathogens have also evolved many strategies to avoid, adapt and even hijack the immune system [4,5]. Among them, antibody-dependent enhancement (ADE) of infection has been well known for kinds of viruses, including HIV, DENV, Zika Virus and adenovirus. ADE phenomenon is thought to occur when viruses are bound by non-neutralizing antibodies or sub-neutralizing concentrations of antibodies that facilitate cell entry in an Fc receptor (FcR)-dependent manner [9]. Furthermore, using adenovirus (Ad) as a model, we reported in this study that preexisting Ad-specific T cell responses contribute to promoting adenovirus infection, suggesting that the virus might also exploit T cell-mediated immune responses to enhance its infection.

Adenovirus-specific T lymphocytes are to be readily activated in response to virus re-infection, and then secret anti-viral cytokines/chemokines (e.g. IFN-γ, TNF-α, IL-2 and granzyme B) to inhibit virus replication or kill virus-infected cells. In contrast, we found that these Ad-specific T lymphocytes also played a role in enhancing Ad infection of monocytes. Our further study demonstrated that this enhancement of Ad infection might be related to the elevated GM-CSF and IL-4 production, which promoted the expression of SR-A and integrin receptors on monocytes. In this way, preexisting Ad-specific T cells in Ad- seropositive subjects might be one of the reasons why CD14+ monocytes from Ad seropositive subjects became more susceptible to Ad infection.

GM-CSF is a well-known cytokine secreted by macrophages, T cells, endothelial cells and fibroblasts in response to immune stimulation. Macrophages or dendritic cells-derived GM-CSF is critical for functional maintenance of antigen-presenting cell (APC) and sequential activation of T cells, while T cell-derived GM-CSF can further enhance the functions of DCs and T cells [29]. GM-CSF has been reported to facilitate the maturation and activation of immune cells through STAT3 and STAT5, and promote host defense against cancer and infections [30]. In clinic, recombinant GM-CSF has being extensively used for treating a variety of cancers. For example, the oncolytic herpes simplex virus type 1 (HSV-1) secreting GM-CSF cytokines had been approved to therapy advanced malignant melanoma by USA FDA [31,32,33]. However, we reported here that incubation of GM-CSF significantly increased the efficacy of adenovirus infection through upregulating the SR-A and integrin receptor. Actually, one previous study also observed that exposure of monocytes to GM-CSF and M-CSF rendered these cells susceptible to adenovirus-mediated gene delivery [34]. Considering that GM-CSF has being administrated widely for clinic use, this observation should be significant in uncovering a potential public health issue.

Interleukin 4 (IL-4) is a cytokine that has many biological functions, such as inducing differentiation of Th0 cells to Th2 cell, and stimulating activation and proliferation of T cell and B cell. IL-4 is usually secreted by activated Th2 cells. Interestingly, we observed for the first time that incubation with recombinant IL-4 can increase adenovirus infection through upregulating the expression of SR-A and integrin receptor on CD14+ monocytes. This is consistent with a recent report that IL-4 stimulation enhanced T-cell activation and increased the replication of dengue virus in CD14+ dermal dendritic cells (dDCs), mainly through up-regulating virus-binding lectins Dendritic Cell–Specific Intercellular adhesion molecule-3–Grabbing Nonintegrin (DC-SIGN/CD209) and mannose receptor (CD206) [35]. Also, increased IL-4 production was correlated with disease progression of HIV-1 patients [36]. Thus, these data suggests that IL-4 could be playing a crucial role in mediating the pathogenesis of a variety of viruses, such as dengue virus, HIV-1 and adenovirus. Furthermore, these studies also implied the possibility that the efficacy of kinds of viral infection and viral-based vector might be regulated by IL-4 administration. For example, blockade of IL-4 signal pathway may be a novel strategy to control viral infections. In contrast, stimulation of IL-4 signal pathway could be a promising strategy to enhance the efficacy of Ad-vectored products.

As a central coordinator of immune system, IFN-γ is a well-studied pleiotropic cytokine with antiviral, antitumor and immunomodulatory properties [37,38], and previous studies showed that IFN-γ inhibited the replication and late transcription of adenovirus [39,40]. Consistent with these findings, we observed that incubation of recombinant IFN-γ protein significantly decreased the efficacy of adenovirus infection in this study, while increased expression of SR-A and integrins on CD14+ cells was seen post IFN-γ administration, potentially causing macrophages and DCs more susceptible to adenovirus infection, which confirms previous observation that the SR-A receptor was up-regulated by IFN-γ administration on the surface of mouse macrophage cell line raw264.7 and human THP-1 monocytes and blood monocytes as well [41,42]. Therefore, these data suggest that IFN-γ might play dual roles during adenovirus infection, counteracting between the virus-inhibiting ability and virus-promoting function mediated by receptor upregulation.

Adenoviruses-based vectors, especially human adenovirus serotype 2 and serotype 5 (Ad2, Ad5), have been extensively developed as gene delivery vehicles for gene therapy and vaccination. Targeting adenovirus-vectored vaccine to dendritic cell (DC) is an effective strategy to enhance immune efficiency, since DC is a highly specialized subset of APC that play a critical role in priming and regulating innate and adaptive immune responses [43]. Cell-based immunotherapies, in which the procedures are usually a pulse of autologous DCs or PBMCs with Ad-vectored products in vitro followed by infusion back into bodies, have been widely explored for treating cancer and infectious diseases [24,44,45]. Coxsackievirus and Adenovirus Receptor (CAR) and integrins play a major role in adenovirus entry into host cells. However, previous studies indicated that human DCs and PBMCs expressed very low levels of these receptors on their cell surfaces and therefore were poorly susceptible to adenovirus-mediated gene delivery [46]. To increase the efficacy of viral entry into targeted cells, some studies performed the infection procedures under centrifugation [47,48]. This method was effective to enhance viral infection efficacy, but it is not widely applicable in clinic due to potential adverse effects on cell viability, time consuming, and complicated medical facility for handling the procedure in the laboratory [49,50]. In this study, our data demonstrated that incubation of IL-4 and GM-CSF enhanced Ad-mediated gene delivery by up-regulating SR-A and integrins αVβ5 receptor. That is, IL-4 and/or GM-CSF-treated DCs and PBMCs will be more susceptible to Ad-mediated gene delivery. Therefore, we suggested an alternative method using the recombinant IL-4 and GM-CSF to improve the efficacy of Ad-vectored gene therapy and vaccination.

For safety in clinic administration, recombinant adenoviral vector as gene delivery vehicles for gene therapy are usually replication-defective due to E1 gene deletion. Consistent with this issue, we used the replication-defective HAdV carrying reporter genes in this study. It is rational to use these viruses, since the aim for this study is to address how the efficacy of adenovirus infection is influenced by the presence of the preexisting virus-specific T lymphocytes. The further study can be performed to know how viruses will replicate in the targeted cells by using replicating adenovirus.

Overall, this study demonstrates that there are complicated interactions between adenovirus and host immune system during long-term co-evolution (Figure 6). Briefly, adenovirus infection triggers potent innate immunity and adaptive immune responses, including B lymphocytes-mediated humoral immunity and T lymphocytes-mediated cellular immunity. Antibodies, which are secreted by B plasma cells, play dual roles in both inhibiting (by neutralizing virus entry) and promoting infection (by ADE). On the other side, antigen-specific T lymphocytes also play dual roles in both inhibiting (by CTL) and promoting infection (by cytokine-dependent enhancement, CDE). Nevertheless, whether the dual roles of T cells in inhibiting and enhancing infection also exist for other viral infections remain elusive and should be further investigated. These findings provide a novel explanation to understand the pathogenesis of viral infections, and give rise to further clarify more subtle counteractions between virus and immune system.

## Figures and Tables

**Figure 1 viruses-11-00154-f001:**
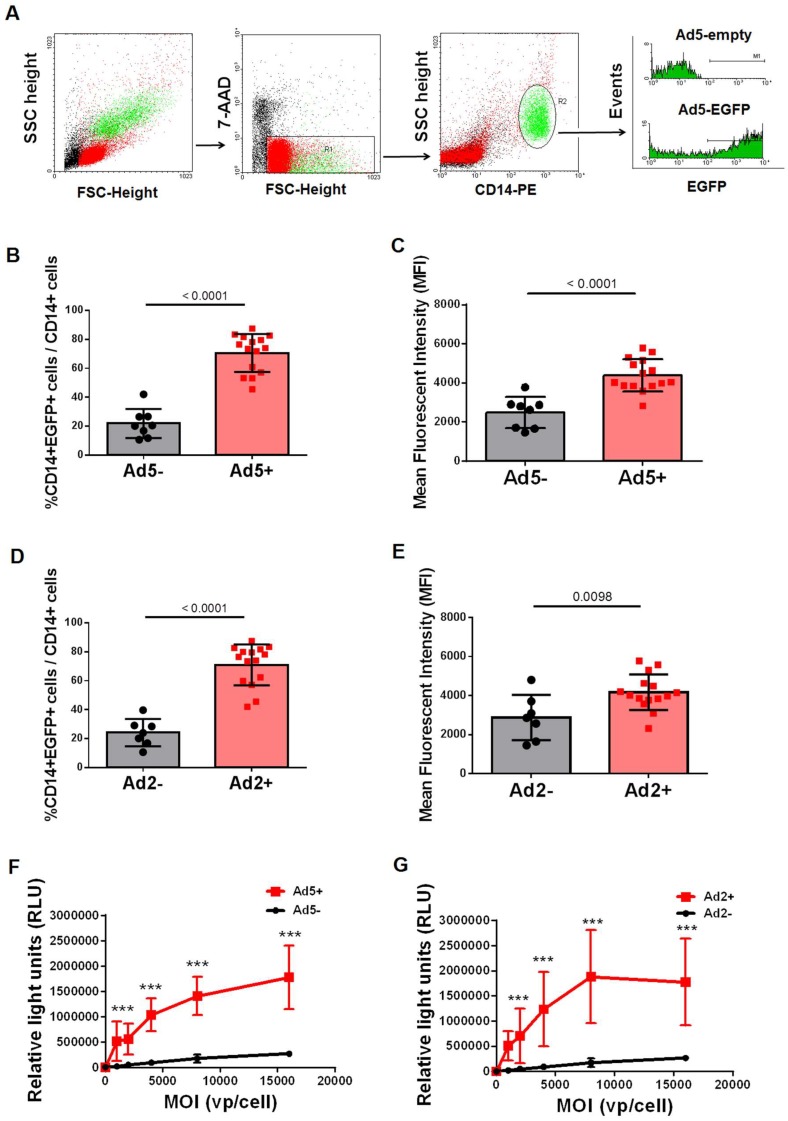
CD14+ monocytes from Ad-seropositive individuals exhibited an increased susceptibility to Ad infection. (**A**) Representative graphics of FACS analysis. Human PBMCs were infected with Ad5-EGFP or Ad2-EGFP, and the proportions of EGFP-positive cells in different cell populations were determined by FACS analysis. Herein, EGFP-positive cells represent those cells that have been infected with Ad5 or Ad2. (**B**) Percentage of EGFP-positive cells in CD14+ cells of Ad seropositive subjects and seronegative subjects after infected with Ad5-EGFP at 1250 vp/cell. (*n* = 6). (**C**) Expression level of EGFP protein (represented with MFI value) in CD14+ cells from Ad seropositive subjects and seronegative subjects after infected with Ad5-EGFP at 1250 vp/cell (*n* = 6). (**D**) Percentage of EGFP-positive cells in CD14+ cells of Ad seropositive subjects and seronegative subjects after infected with Ad2-EGFP at 1250 vp/cell (*n* = 6).(**E**) Expression level of EGFP protein (represented with MFI value) in CD14+ cells from Ad seropositive subjects and seronegative subjects after infected with Ad2-EGFP at 1250 vp/cell (*n* = 6). (**F**) Percentage of RLU (Relative light units) in PBMC of Ad seropositive subjects and seronegative subjects after infected with Ad5-SEAP at 0, 1000, 2000, 4000, 8000, and 16,000 vp/cell. (*n* = 6). (**G**) Percentage of RLU (Relative light units) in PBMC of Ad seropositive subjects and seronegative subjects after infected with Ad2-SEAP at 0, 1000, 2000, 4000, 8000, and 16,000 vp/cell. (*n* = 6). The bars represent the standard error. ***: *p* < 0.001.

**Figure 2 viruses-11-00154-f002:**
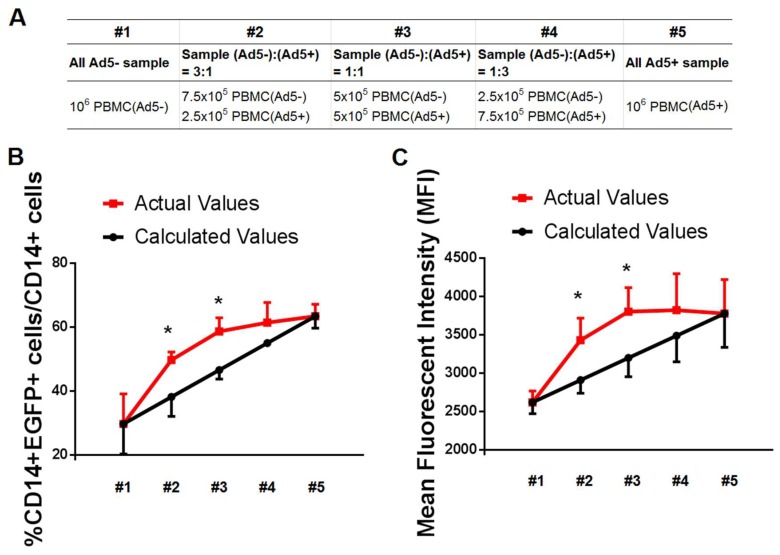
Secretory factors by PBMC samples of Ad-seropositive individuals contributed to promoting the efficacy of Ad infection. (**A**) Representative of paired PBMCs from Ad5-seronegative subjects and Ad5-seropositive subjects (*n* = 3 pairs) mixed together at different ratios. (**B**) The actual and calculated percentage of EGFP-positive cells in CD14+ cells of PBMC mixture after infected with 1250 vp/cell Ad5-EGFP. (**C**) The actual and calculated expression level of EGFP protein (represented with MFI value) in CD14+ cells of PBMC mixture after infected with 1250 vp/cell Ad5-EGFP. These data were analyzed statistically by nonparametric Mann-Whitney U test. The bars represent the standard error. *: *p* < 0.05.

**Figure 3 viruses-11-00154-f003:**
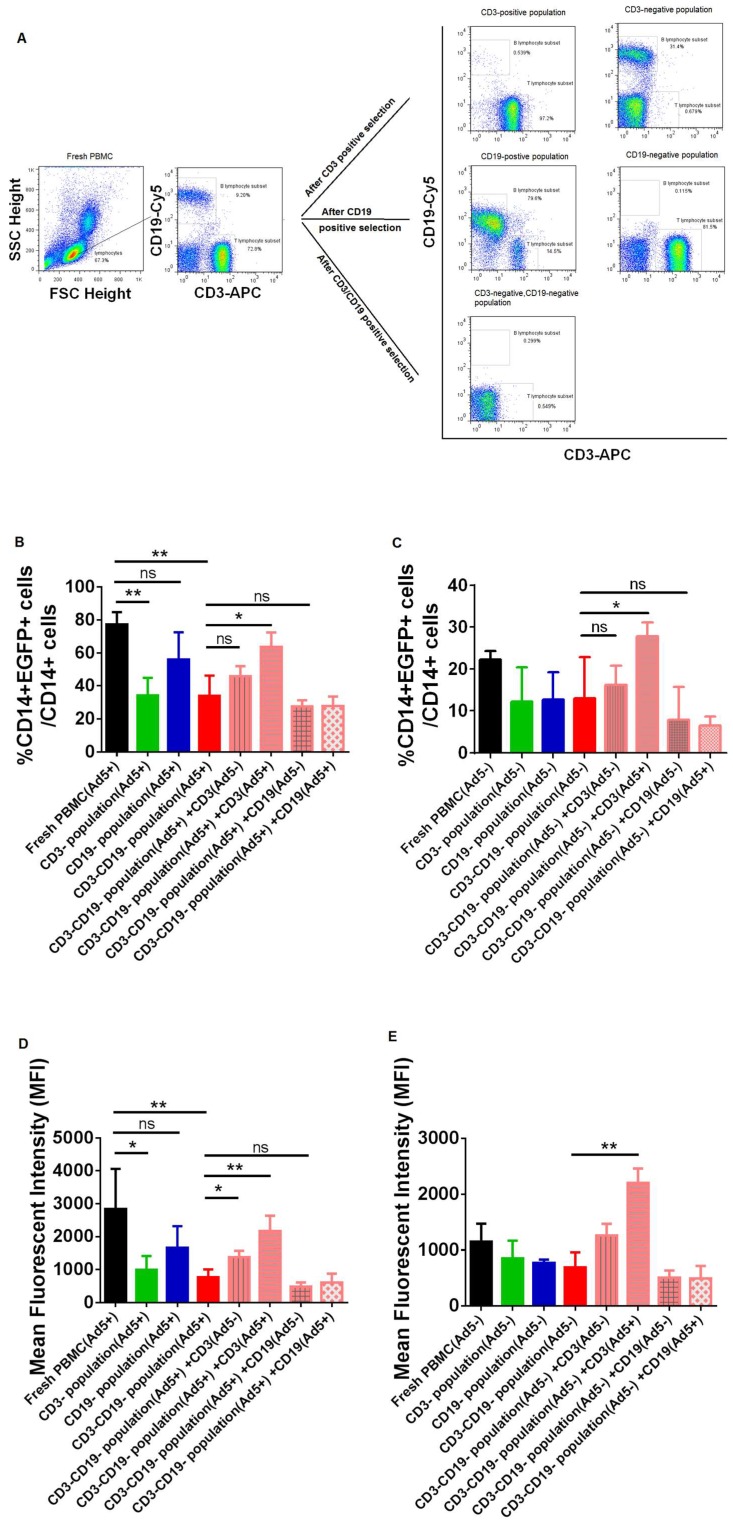
The efficacy of adenovirus infection into CD14+ monocytes was significantly affected by removing or adding CD3+ T lymphocytes from PBMC samples of Ad-seropositive individuals. Different cell populations were magnetically sorted from human PBMCs as described in methods, and then infected with 1250vp/cell of Ad5-EGFP. After incubation for 24 h, the efficacy of Ad infection was detected. (**A**) Representative graphics of FACS analysis and gating strategy to determine the purity of sorted cell populations. (**B**) Percentage of EGFP positive cells in CD14+ monocytes from Ad5-seropositive samples (*n* = 3) under different treatments, including fresh PBMCs, CD3-depleted PBMCs [CD3- population (Ad5+)], CD19-depleted PBMCs [CD19- population(Ad5+)], CD3-/CD19- double depleted PBMCs [CD3-CD19- population], CD3-/CD19- double depleted PBMCs but suppled with CD3+T cells from Ad5-seronegative subjects [CD3-CD19- population(Ad5+)+CD3(Ad5-)], CD3-/CD19- double depleted PBMCs but suppled with CD3+T cells from Ad5-seropositive subjects [CD3-CD19- population(Ad5+)+CD3(Ad5+)], CD3-/CD19- double depleted PBMCs but suppled with CD19+ B cells from Ad5-seronegative subjects [CD3-CD19- population(Ad5+)+CD19(Ad5-)], and CD3-/CD19- double depleted PBMCs but suppled with CD19+ B cells from Ad5-seropositive subjects [CD3-CD19- population(Ad5+)+CD19(Ad5+)]. (**C**) Percentage of EGFP positive cells in CD14+ monocytes from Ad5-seronegative samples (*n* = 3) under different treatments as descripted above. (**D**) Expression level of EGFP protein (represented with MFI value) in CD14+ monocytes from Ad-seropositive subjects (*n* = 3) as descripted above. (**E**) Expression level of EGFP protein (represented with MFI value) in CD14+ monocytes from Ad-seronegative subjects (*n* = 3) as descripted above. The bars represent the standard error. *: *p* < 0.05, **: *p* < 0.01, ns: no significance.

**Figure 4 viruses-11-00154-f004:**
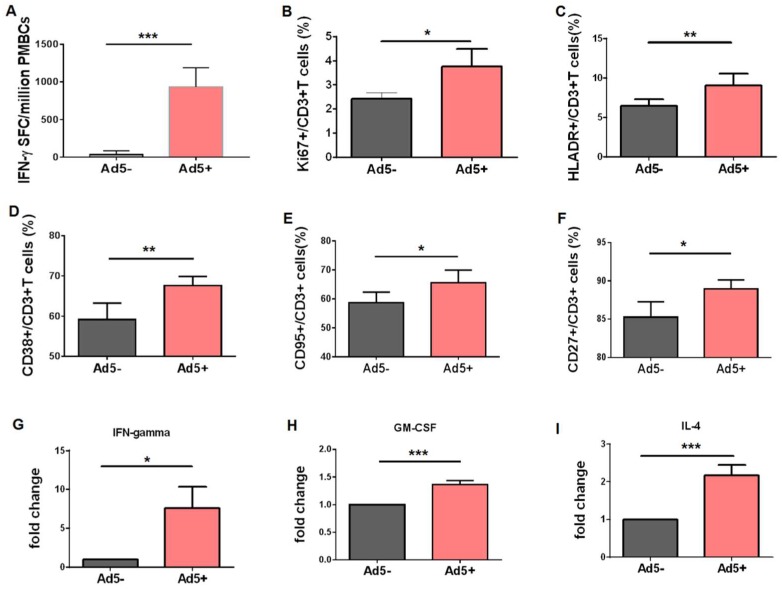
Pre-existing virus-specific CD3+ T lymphocytes were more effectively activated after encountering adenovirus stimulus. (**A**) PBMCs of Ad-seropositive subjects (*n* = 15) or seronegative subjects (*n* = 5) were isolated, and the level of Ad-specific preexisting T lymphocytes in PBMCs was assessed ELISPOT assay as described in materials. (**B**–**F**) PBMCs of Ad-seropositive subjects (*n* = 3) or seronegative subjects (*n* = 3) were infected with 1250 vp/cell of recombinant Ad virus for 48 h, and the levels of cell proliferation (**B**), activation state including HLA-DR (**C**), CD38 (**D**), CD95 (**E**), CD27 (**F**), and cytokines/chemokines expression including IFN-γ (**G**), GM-CSF (**H**) and IL-4 (**I**) were further assessed as described in materials. The bars represent the standard error. *: *p* < 0.05, **: *p* < 0.01, ***: *p* < 0.001.

**Figure 5 viruses-11-00154-f005:**
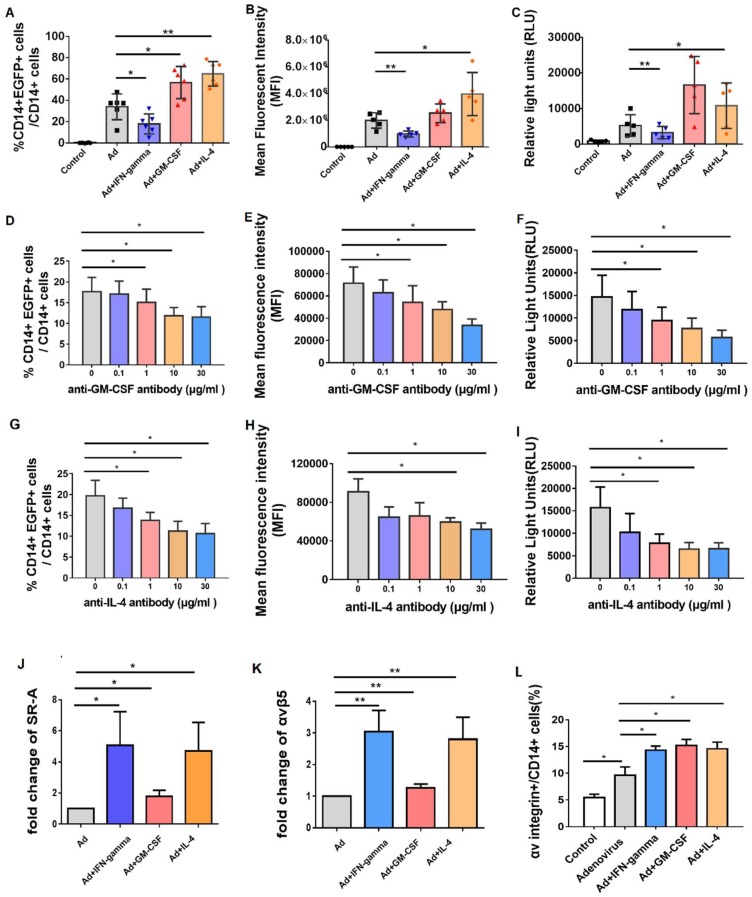
Incubation of GM-CSF and IL-4 promoted the efficacy of adenovirus infection by upregulating SR-A and integrin receptor. 1250 vp/cell of Ad5-EGFP or Ad5-SEAP virus was added into PBMCs with or without recombinant IFN-γ (40 ng/mL), GM-CSF (20 ng/mL) or IL-4 (20 ng/mL) respectively, and the expression of EGFP or SEAP was detected after 48 h. (**A**) The change of percentage of EGFP-positive cells in the different cytokines-treated CD14+ cells (*n* = 6). (**B**) The change of expression level of EGFP protein (MFI value) in the different cytokines-treated CD14+ cells (*n* = 6). (**C**) The change of RLU (Relative light units) in the different cytokines-treated PBMCs (*n* = 6). 1250 vp/cell of Ad5-EGFP or Ad5-SEAP virus was added into PBMCs (*n* = 3) with isotype control antibody or anti-human GM-CSF monoclonal antibody (0–30 μg/mL), and the percentage of EGFP-positive cells (**D**), the expression level of EGFP protein (MFI value) (**E**) and the expression level of SEAP (RLU value) (**F**) were detected after 24 h. 1250 vp/cell of Ad5-EGFP or Ad5-SEAP virus was added into PBMCs (*n* = 3) with isotype control antibody or anti-human IL-4 monoclonal antibody (0–30 μg/mL), and the percentage of EGFP-positive cells (**G**), the expression level of EGFP protein (MFI value) (**H**) and the expression level of SEAP (RLU value) (**I**) were detected after 24 h. Moreover, the expression level of adenovirus-related receptors, including SR-A (**J**) and αV integrin (**K**) was detected by quantitative RT-PCR. The integrin receptor protein on the cell surface was also analyzed by flow cytometer (**L**). The bars represent the standard error. *: *p* < 0.05, **: *p* < 0.01.

**Figure 6 viruses-11-00154-f006:**
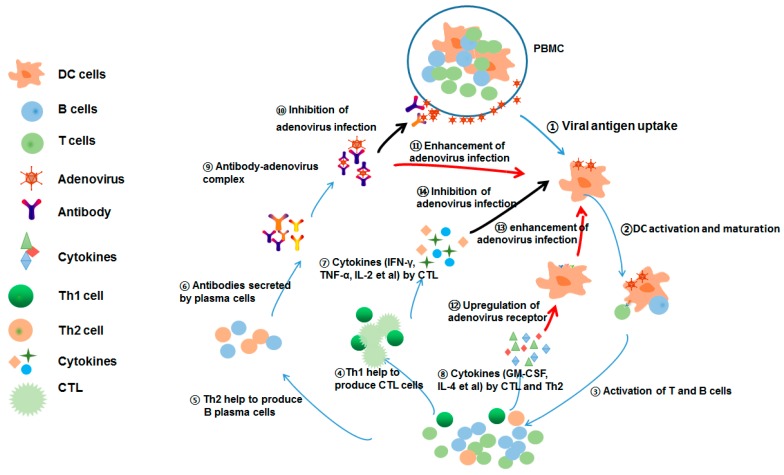
Pattern to illustrate the interactions between adenoviral infection and T/B lymphocytes. Adenovirus was captured and uptake by APCs (monocyte, macrophages and DCs), and this triggered those APCs to activate and mature following presenting degraded adenoviral peptides to stimulate T lymphocytes and B lymphocytes in turn. Subsequently, adenovirus-specific T and B lymphocytes were elicited to proliferate and differentiate into the effector and memory immune cells. B lymphocyte can further differentiate into plasma cells to secrete antibodies with the help of Th2 cells. The antibodies usually neutralize and block the entry of adenovirus into host cells. However, adenovirus also binds to those antibodies to form antibody-virus complex to enhance their infectivity to host cells by an Fc receptor (FcR)-dependent manner, which is termed as antibody-dependent enhancement (ADE). On the other side, adenovirus-specific T lymphocytes are rapidly activated to secrete some cytokines/chemokines (IFN-γ, TNF-α, IL-2 and granzyme B et al) to inhibit viral replication and eradicate the virus-infected cells. However, the increased cytokines of GM-CSF and IL-4 by T lymphocytes can promote adenovirus infection by up-regulating scavenger receptor 1 (SR-A) and integrins αVβ5 receptor, and we termed this phenomenon as cytokine-dependent enhancement (CDE). Taken together, after long-term co-evolution, there are complicated interactions between adenovirus and host immune system.

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
