# Peer review of "Preexisting Virus-Specific T Lymphocytes-Mediated Enhancement of Adenovirus Infections to Human Blood CD14+ Cells"

_viruses, 2019, doi:10.3390/v11020154_

Reviewer 1 Report

I am happy that my previous comments have been addressed and additional experiments performed to clarify mechanistic aspects of viral enhancement. 

There is one outstanding point to consider - For the cytokine blocking experiment using neutralising monoclonal antibodies, did the authors include an isotype-matched control reagent? If so this should be included or stated in the methods/ text.

Author Response

Response: Thanks for your comments. We used the matched isotype control antibody as mock control in our study, and we have stated it in revised methods/text (Line 164). Thank you.

Reviewer 2 Report

no comments

Author Response

Thank you for reviewing our work.

Reviewer 3 Report

The study by Feng et al., has focused on understanding how HAdV-2/5 infections can be influenced by the presence of the preexisting virus-specific T lymphocytes. Further, this elegant study reveals that the GM-CSF and IL-4 cytokines can specifically enhance HAdV infections via promoting the expression of the specific cell receptors ( e.g., SR-A).

The study is well-conducted, the results are well-presents and finally the discussion is nicely balanced between the author's results, literature and potential practical usage of the research in the clinics.

I have only a few minor comments:

Line 93: "...rhesus monkey,...." I could not find a single experiment where the monkey PBMCs were used. In the case these cells were used, please state where as well as why the rhesus monkey cells were used instead of the human PBMCs. Also please provide the ethical permission for rhesus monkey PBMCs, incl. the reference code.

Line 98: provide reference number for ethical permission

Line 161: it is difficult to believe that 0.2-1 million cells were seeded per well in 96-well plate. Please correct it if wrong or convince me that it is possible.

More general comment.

So, the authors use reporter viruses (EGFP, SEAP) throughout the study to show the infectivity of the PBMCs, CD14+ monocytes, or CD3+ T lymphocytes. This is all fine, but it would have been interesting to see/ compare how HAdV will replicate in these target cells. Of course this will need replicating virus infections and could not be done with the viruses used by the authors.Further, it will be not the focus of the manuscript either.  Anyhow, I would urge the authors to add a comment, (e.g., based on their own research or available literature) how HAdV-2/5 would replicate (and not only infect) in their studied cells.This will help the reader to better understand the virus infectious cycle in the PBMCs.

Author Response

Line 93: "...rhesus monkey,...." I could not find a single experiment where the monkey PBMCs were used. In the case these cells were used, please state where as well as why the rhesus monkey cells were used instead of the human PBMCs. Also please provide the ethical permission for rhesus monkey PBMCs, incl. the reference code.

Response: Thanks for your careful reading. We have deleted this typo from Line 93. Thanks.

Line 98: provide reference number for ethical permission

Response:  We have added the permit number in revised text (Line 97).

Line 161: it is difficult to believe that 0.2-1 million cells were seeded per well in 96-well plate. Please correct it if wrong or convince me that it is possible.

Response: Thanks for your careful reading. We have corrected this sentence as “PBMC was seeded at 2x105 per well in 96-well plates or 1x106 cells per well in 24-well plates”. By the way, PBMC cells are suspended and do not adhere to the bottom of the culture plate, so there is no problem to culture 1 million PBMC cells per well in 24-well plate or 96-well plate. Thank you very much.

More general comment. So, the authors use reporter viruses (EGFP, SEAP) throughout the study to show the infectivity of the PBMCs, CD14+ monocytes, or CD3+ T lymphocytes. This is all fine, but it would have been interesting to see/ compare how HAdV will replicate in these target cells. Of course this will need replicating virus infections and could not be done with the viruses used by the authors.Further, it will be not the focus of the manuscript either.  Anyhow, I would urge the authors to add a comment, (e.g., based on their own research or available literature) how HAdV-2/5 would replicate (and not only infect) in their studied cells.This will help the reader to better understand the virus infectious cycle in the PBMCs.

Response: Thanks for your comments. We add a comment in the revised manuscript ( line 439-444). “For safety in clinic administration, recombinant adenoviral vector as gene delivery vehicles for gene therapy are usually replication-defective due to E1 gene deletion. Consistent with this issue, we used the replication-defective HAdV carrying reporter genes in this study. It is rational to use these viruses, since the aim for this study is to address how the efficacy of adenovirus infection is influenced by the presence of the preexisting virus-specific T lymphocytes. The further study can be performed to know how viruses will replicate in the targeted cells by using replicating adenovirus.” Thank you.

This manuscript is a resubmission of an earlier submission. The following is a list of the peer review reports and author responses from that submission.

Round  1

Reviewer 1 Report

This paper provides evidence that T cell derived factors could contribute to enhanced infectivity of human monocytes by Adenovirus. The experiments are well performed and presented and the data are mostly convincing and supportive of the conclusions.

There are some points which should be addressed. One straightforward experiment should be performed (see point 3, below).

1.      I am uncertain as to the merits of the experiment described in Figure 2 – it is clear that the MLR is not representative of the normal situation for adenovirus (re)infection.

The purpose of the experiment requires clearer explanation:

-The hypothesis that T cell derived factors stimulated in MLR could support adenovirus infection should be more clearly stated.

- I am assuming that 3 pairs of Ad5+ and Ad5- donors are used at each dilution ratio - please state this clearly.

2.       The data in the following figure3  are more straight forward to  interpret. The authors should however present data (in the materials and methods or as supplementary data) detailing the mean and range of percentages of contaminating T and B cells for the group of individuals shown in Figures 3B to 3E. The data for reconstitution with T cells from Ad5+ versus Ad5- individuals demonstrate a significant recovery of infectivity with the former. This does not conclusively demonstrate a role for Ad5 specific T cells -  there is some partial recovery with Ad5- T cells although not significant. The authors should also address the caveat that positive selection of T cells via CD3 microbeads could lead to activation. Have the authors tried depleting CD4+ T cells?

Please state the number of individuals used in figure 3B to 3E  in the text.

3.    A. Figure 4 and 5 do provide evidence that factors derived from Ad -specific T cells could provide factors which support enhanced adenovirus infection of monocytes. IFN-g would be expected to have a protective, limiting effect of virus infectivity, supported by the data in figure 5A-C and the model in figure 6. It is striking however that changes in IFN-g mRNA, are greater than both GMCSF and IL-4.  Is there any correlation between the levels of these factors and the level of virus infectivity?

B. More importantly, this series of experiments should be concluded more conclusively by using neutralising monoclonal antibodies (with isotype matched control reagents to block IL-4, GM-CSF or both of these factors in PBMC from a number of Ad5+ compared to Ad5- individuals.

4.      There are some minor points of the text style which could be improved. For example the words ‘in general’ should be omitted from the beginning of the abstract as this does not reflect the undoubtable role of CD8+ T cells in protection against virial infections.

Author Response

1.      I am uncertain as to the merits of the experiment described in Figure 2 – it is clear that the MLR is not representative of the normal situation for adenovirus (re)infection.

The purpose of the experiment requires clearer explanation:

-The hypothesis that T cell derived factors stimulated in MLR could support adenovirus infection should be more clearly stated.

- I am assuming that 3 pairs of Ad5+ and Ad5- donors are used at each dilution ratio - please state this clearly.

Answer: Thanks for your mentions. In our previous studies, we observed that CD14+ cells from Ad5-seropositive subjects exhibited an increased susceptibility to Ad5 entry, when compared with that of Ad5-seronegative subjects. The increased receptors (such as integrin and SR-A) on the surface of targeted cells might be partially associated with this observation, but the underlying mechanism is not fully understood. To further elucidate if there are other factors (such as secretory cytokines by activated lymphocytes) to affect the efficacy of Ad infection, we performed this experiment of mixed lymphocyte culture. We speculated if there were no other secretory factors to affect the efficacy of Ad infection, the predicted values of the different ratios of PBMCs mixture were expected to be a linear relation. However, the actual values of infection efficacy in our study were a nonlinear curve relation (Figure 2B), implying that the secretory cytokines by activated lymphocytes did affect the efficacy of Ad infection. Following your suggestions, we have stated clearly the purpose of this experiment in revised manuscript (line 209-220). Thank you.

2.       The data in the following figure3 are more straight forward to interpret. The authors should however present data (in the materials and methods or as supplementary data) detailing the mean and range of percentages of contaminating T and B cells for the group of individuals shown in Figures 3B to 3E. The data for reconstitution with T cells from Ad5+ versus Ad5- individuals demonstrate a significant recovery of infectivity with the former. This does not conclusively demonstrate a role for Ad5 specific T cells -  there is some partial recovery with Ad5- T cells although not significant. The authors should also address the caveat that positive selection of T cells via CD3 microbeads could lead to activation. Have the authors tried depleting CD4+ T cells?

Please state the number of individuals used in figure 3B to 3E in the text.

Answer: Thanks for your comment. According to your suggestions, we added the supplementary Table 2 as supporting data. We have added the description more clearly in revised manuscript (line 263-269). “Then, the purity of sorted cells and the percentage of residue contaminating cells was determined by FACS (Figure 3A and supplementary Table 2).  In this study, the purity of sorted CD3+ T cells and CD19+ B cells was 90.02%-97.2% and 76.23%-80.02% respectively. The residue contaminating B cells in sorted CD3+ T cells was 0.54%-1.63%, and the residue contaminating T cells in sorted CD19+ B cells was 11.10%-15.02%. The percentage of CD19+ B cells in sorted CD3-/CD19- sample was 0.23%-0.55%, and the percentage of CD3+ T cells in sorted CD3-/CD19- sample was 0.11%-0.38%.”

We have addressed the concern of the partial restoring by Ad5-T cells and the cell activation by the positive selection via microbeads in revised manuscript (line 271-275). “In our study, adding CD3+ T lymphocytes from Ad-seronegative individuals (non-specific CD3+ T cells) did not significantly enhance the efficacy of Ad infection, although there is a trendy of partially restored. This partial restoring might be caused by the non-specific cell activation by the positive selection via magnetic microbeads.” 

In this study, we want to know how the whole populations of T lymphocytes or B lymphocytes affect the efficacy of adenovirus infection into CD14+ cells, so we did not try to sort the different subpopulations of CD4+T cells, CD8+ T cells etc. Thanks for your suggestion, we will try this issue in another project.

We added the number of samples used in revised Figure 3 legend. Thanks.

3.    A. Figure 4 and 5 do provide evidence that factors derived from Ad -specific T cells could provide factors which support enhanced adenovirus infection of monocytes. IFN-g would be expected to have a protective, limiting effect of virus infectivity, supported by the data in figure 5A-C and the model in figure 6. It is striking however that changes in IFN-g mRNA, are greater than both GMCSF and IL-4.  Is there any correlation between the levels of these factors and the level of virus infectivity?

B. More importantly, this series of experiments should be concluded more conclusively by using neutralising monoclonal antibodies (with isotype matched control reagents to block IL-4, GM-CSF or both of these factors in PBMC from a number of Ad5+ compared to Ad5- individuals.

Answer: Thanks for your suggestions. To further verify the roles of GM-CSF and IL-4 in promoting adenovirus infection, we administrated the neutralizing antibodies to block the cytokines-dependent enhancement of viral infection. As expected, both the frequency of EGFP-positive cells (Figure 5D) and expression level of EGFP protein (MFI, Figure 5E) were significantly decreased in CD14+ cells of anti-human GM-CSF monoclonal antibody-treated samples in a dose-dependent manner, when comparted with mock-treated samples. Moreover, after Ad-SEAP incubation with anti-human GM-CSF monoclonal antibody simultaneously, the expression level of SEAP was significantly reduced in a dose-dependent manner (Figure 5F). In addition, the similar results were found when Ad-EGFP or Ad-SEAP was incubated with anti-human IL-4 monoclonal antibody simultaneously (Figure 5G,5H,5I).

We have added the method and results of this new experiment in revised manuscript (line 162-169, line 331-336 and line 340-348). Thank you.

4.      There are some minor points of the text style which could be improved. For example, the words ‘in general’ should be omitted from the beginning of the abstract as this does not reflect the undoubtable role of CD8+ T cells in protection against virial infections.

Answer: Thanks for your mention. We have carefully checked the use of English within this manuscript.

Reviewer 2 Report

The study by Feng and colleagues proposes that donors with pre-existing humoral immunity to adenoviruses have T  lymphocytes that enhance infection of adenoviruses (specifically Ad2 & 5). 

There several major major flaws that prevents me from being positive about this study.

1) One cannot speak globally about pre-existing Ad humoral  immunity, especially when using a NAb assay. NB >95 of all individuals have anti-Ad Abs, some of theses are type specific.

2) One cannot  create a link between Ad2 or Ad5 NAb titres and the level of Ad-specific cellular immunity. Cellular immunity is cross-reactive, NAb titre are (generally) not cross-reactive. Unless each donor is characterized for his/her level of T cell immunity all the information in this study worth nothing.  

3) I am unaware of any other group supporting this line of data, in spite of the 3rd line on page 9. 

4) There is no "mechanistic" explanation or assay anywhere in this study. It is a collection of observations that do not justify the conclusions.

Author Response

1) One cannot speak globally about pre-existing Ad humoral immunity, especially when using a NAb assay. NB >95 of all individuals have anti-Ad Abs, some of theses are type specific.

Answer: Thanks for your question. But it is well known that a major obstacle for clinical application of Ad5-vectored products is that populations worldwide have a high prevalence of anti-Ad5 neutralizing antibodies, implying extensively prior natural infection. Actually, both our previous data and others studies showed that approximately 40% of the adult population in America (Nwanegbo et al., 2004), 77% of the total population in Guangzhou of China (Sun et al., 2011), and more than 93% of the children in Sub-Saharan Africa (Thorner et al., 2006) are Ad5-seropositive. We have described this situation more clearly in revised manuscript. Thanks.

2) One cannot create a link between Ad2 or Ad5 NAb titres and the level of Ad-specific cellular immunity. Cellular immunity is cross-reactive, NAb titre are (generally) not cross-reactive. Unless each donor is characterized for his/her level of T cell immunity all the information in this study worth nothing.  

Answer: Actually, to investigate the roles of T cellular immunity during adenovirus infection, we detect both anti-Ad Nab titres and Ad-specific cellular immunity for the samples used in this study (Figure 4A). Our data showed that the frequency of Ad5-specific IFN-γ-secreting T lymphocytes was significantly higher than that of T lymphocytes in Ad5-seronegative PBMCs (Figure 4A, p<0.0001), implying that there was high level of preexisting Ad5-specific T memory cells in Ad5-seropositive subjects.

3) I am unaware of any other group supporting this line of data, in spite of the 3rd line on page 9. 

Answer: Basically, this is the first time to report that the preexisting virus-specific T lymphocytes might mediate the enhancement of adenovirus infections. However, some supporting works can be found in the references. For example, one previous study observed that exposure of monocytes to GM-CSF and M-CSF rendered these cells susceptible to adenovirus-mediated gene delivery (Huang et al. 1995).

4) There is no "mechanistic" explanation or assay anywhere in this study. It is a collection of observations that do not justify the conclusions. 

Answer: In fact, there are mechanistic studies in our work. As showed in Figure 5 and Figure 6, we performed several experiments to explain our observation. In brief, adenovirus-specific T lymphocytes are rapidly activated to secrete some cytokines/chemokines (IFN-γ, TNF-α, IL-2 and granzyme B et al) to inhibit viral replication and eradicate the virus-infected cells. However, the increased cytokines of GM-CSF and IL-4 by T lymphocytes can promote adenovirus infection by up-regulating scavenger receptor 1 (SR-A) and integrins αVβ5 receptor. Furthermore, we are doing more experiments to verify this conclusion in vivo animal model in another project. Hopefully, we will report it in another paper. Thanks for your understanding.